# Listen or Read? The Impact of Proficiency and Visual Complexity on Learners’ Reliance on Captions

**DOI:** 10.3390/bs15040542

**Published:** 2025-04-17

**Authors:** Yan Li

**Affiliations:** School of Foreign Languages and Cultures, Jilin University, Changchun 130012, China; y_li20@mails.jlu.edu.cn

**Keywords:** captioned videos, proficiency levels, language processing, desirable difficulties, video complexity, listening skills

## Abstract

This study investigates how Chinese EFL (English as a foreign language) learners of low- and high-proficiency levels allocate attention between captions and audio while watching videos, and how visual complexity (single- vs. multi-speaker content) influences caption reliance. The study employed a novel paused transcription method to assess real-time processing. A total of 64 participants (31 low-proficiency [A1–A2] and 33 high-proficiency [C1–C2] learners) viewed single- and multi-speaker videos with English captions. Misleading captions were inserted to objectively measure reliance on captions versus audio. Results revealed significant proficiency effects: Low-proficiency learners prioritized captions (reading scores > listening, *Z* = −4.55, *p* < 0.001, *r* = 0.82), while high-proficiency learners focused on audio (listening > reading, *Z* = −5.12, *p* < 0.001, *r* = 0.89). Multi-speaker videos amplified caption reliance for low-proficiency learners (*r* = 0.75) and moderately increased reliance for high-proficiency learners (*r* = 0.52). These findings demonstrate that low-proficiency learners rely overwhelmingly on captions during video viewing, while high-proficiency learners integrate multimodal inputs. Notably, increased visual complexity amplifies caption reliance across proficiency levels. Implications are twofold: Pedagogically, educators could design tiered caption removal protocols as skills improve while incorporating adjustable caption opacity tools. Technologically, future research could focus on developing dynamic captioning systems leveraging eye-tracking and AI to adapt to real-time proficiency, optimizing learning experiences. Additionally, video complexity should be calibrated to learners’ proficiency levels.

## 1. Introduction

Videos have become a common and major way to provide input to learners in numerous second language (L2) classrooms for several decades ([3]; [5]; [27]). Many teachers use captioned videos to help improve the learners’ listening abilities, as they expose learners to authentic oral language input that closely resembles real-life communication. However, there is a view that learners may simply read the captions and ignore the soundtrack, thus improving reading rather than listening skills ([25]; [34]). Prior studies yield mixed findings: Low-proficiency learners consistently depend on captions as a crutch ([2]; [28]), while higher-proficiency learners integrate audio, visuals, and text ([23]; [29]). However, contradictory evidence suggests even advanced learners may default to caption reliance ([3]) if the task complexity exceeds their cognitive capacity.

Given the inconclusive and occasionally conflicting outcomes, further research is needed to investigate whether the learners mainly read the captions or actually listen to the videos. For one thing, understanding this dynamic is essential for optimizing the use of captioned videos in language teaching, as it enables educators to provide learners with appropriate video materials and effective instruction. For another, English educators need an efficient, user-friendly, and scientifically validated tool to assess students’ reliance on captions. Such assessment could inform decisions about when to implement learner-appropriate difficulties, like gradually reducing caption support as proficiency increases. This investigation aligns with cognitive psychology’s concept of “desirable difficulties” ([1])—the principle that introducing optimal challenges (e.g., strategic caption removal) can enhance long-term learning.

In this study, I focused on whether learners primarily listened to the audio or read the captions while watching the videos. To accomplish this, paused transcription was used to investigate how much the participants relied on captions. The findings showed that low-proficiency learners predominantly relied on captions as a scaffold for comprehension, while high-proficiency learners demonstrated a greater ability to integrate auditory and textual cues. Additionally, video complexity, such as the presence of multiple speakers, can increase reliance on captions, particularly for low-proficiency learners. The outcomes aim to provide actionable thresholds for implementing desirable difficulties in caption scaffolding, ultimately informing the development of tailored instructional strategies that account for learners’ proficiency trajectories and multimodal processing capacities.

## 2. Literature Review

In understanding how language learners process captioned videos, two prominent cognitive theories provide valuable frameworks: the cognitive theory of multimedia learning ([13]) and the cognitive load theory ([20], [21]). These theories offer insights into how learners utilize different cognitive resources while watching videos with captions, such as processing auditory, visual, and textual information.

### 2.1. Theoretical Framework

#### 2.1.1. Cognitive Theory of Multimedia Learning (CTML)

The cognitive theory of multimedia learning, developed by [13] ([13]), posits that people learn more effectively from words and pictures than from words alone. According to CTML, learning occurs through two distinct channels in the brain: the auditory channel (for processing spoken words) and the visual channel (for processing pictures and text). Mayer’s theory suggests that effective learning happens when information from these two channels is presented in a way that reduces cognitive overload and facilitates the integration of the material into long-term memory.

The cognitive theory of multimedia learning entails three assumptions: the dual-channel assumption (i.e., information is processed through two channels: an auditory–verbal channel and a visual–pictorial channel), the limited capacity assumption (i.e., each channel has a limited capacity to process information), and the active processing assumption (i.e., humans actively engage in cognitive processing to construct mental representations of their experiences through attending to, organizing, and integrating incoming information).

The cognitive theory of multimedia learning emphasizes several principles demonstrated to affect the cognitive processing of information. One of them is the redundancy principle: including redundant information (e.g., presenting spoken text along with on-screen text) can lead to cognitive overload. In a study conducted by [14] ([14]), native English-speaking participants watched a captioned animation explaining lightning while listening to concurrent narration in English. It turned out that they retained less information than those who watched and listened to the same but uncaptioned animation. However, [22] ([22]) claimed that the principles above are “not borne out for second language learners” (p. 52). Foreign language learners might find that captions support their access to multimedia messages. The question is how language learners are different from native speakers in attending to the three channels of sound, image, and captions.

#### 2.1.2. Cognitive Load Theory (CLT)

Developed by [20] ([20], [21]), cognitive load theory (CLT) is a psychological theory that focuses on how the human cognitive system processes information. The theory posits that there are limits to the amount of information individuals can process effectively in their working memory.

Cognitive load refers to the mental effort and resources required to process information. There are three types of cognitive load: (1) Intrinsic cognitive load: This is the inherent complexity of the content being learned. Some subjects or concepts are naturally more complex and thus impose a higher intrinsic cognitive load. (2) Extraneous cognitive load: This is the cognitive load imposed by the instructional design, presentation, or learning environment. Poorly designed materials can lead to extraneous cognitive load, distracting learners from the main content and impeding learning. (3) Germane cognitive load: This type of cognitive load is directed toward building mental connections between new information and existing knowledge.

Working memory is one of the key concepts on which CLT is built. It is the part of the cognitive system responsible for temporarily holding and manipulating information. It can only process a limited amount of information at a time. When too many messages are presented, selection, organization, and integration occur.

The goal of CLT is to optimize learning by managing cognitive load. In multimedia presentations, teachers can find and expose learners to materials that consider the inherent complexity of subjects, the sequencing of topics, and the use of appropriate instructional strategies to manage cognitive load, ensuring that the materials do not overwhelm learners with excessive cognitive demands.

### 2.2. Empirical Evidence About How Learners Process Captioned Videos

The existing literature on captions focuses mainly on their benefits for second language learning, particularly in two main areas: general comprehension and vocabulary acquisition. Some studies have examined how captioned videos improve learners’ listening comprehension skills. However, only nine studies focus on how learners process captioned videos.

#### 2.2.1. Overview of the Captioned Video Processing

Of the nine studies outlined in the Appendix A, six focused on English learning. Three studies investigating foreign language learning (Russian and Spanish) by native English speakers were also included, as they provided valuable insights and inspiration for this research. Among these, six were cross-sectional, as they collected data on viewing behaviors at a single point in time ([4]). The remaining three ([26], [27], [28]) were longitudinal, as they collected data repeatedly from the same participants over periods of 9 weeks, 5–6 weeks, and 12 weeks, respectively.

Of the six cross-sectional studies, three included low-proficiency learners ([2]; [22]; [29]), two were composed of mixed lower- and higher-proficiency learners ([18]; [23]), and the sixth one targeted upper-intermediate learners. The learners involved in those three longitudinal studies were all at an upper-intermediate to advanced level. Participants in [3]’s ([3]) study were 10 Chinese ESL learners. One Cantonese-speaking student participated in [22]’s ([22]) study. Participants in other studies were mainly European foreign language speakers.

As to the videos used in the nine studies investigated here, seven used videos of various types and lengths, such as comedy series, movies, TV shows, and documentary clips. The exception was [23] ([23]), who used a segment from the videotape included within a textbook, while [2] ([2]) did not provide details about the specific kinds of videos that were utilized. While selecting captioned videos, the researchers of most studies, excluding Caimi, judged the selected videos suitable for the level of the students in their studies through various methods, such as vocabulary tests, instructors’ evaluation, standard tests, etc.

Concerning the research techniques used to gather the data (see the Appendix A), most studies used introspective methods to assess participants’ perceived attention distribution, usually immediately after the viewing task. These techniques included interviews, questionnaires, verbal reports, learner written reflective protocols, and learner diaries (hard-copy or online). [29] ([29]) also used eye tracking, via a sensor-based device which can tap into the learners’ real-time processing of the videos and objectively record participants’ attention distribution between captions and the visuals.

#### 2.2.2. Main Results from the Cross-Sectional Studies

Regarding the findings of the cross-sectional studies, two main results were observed: First, low-proficiency learners mainly read captions, using captions as a scaffold to help them understand the videos; second, high-proficiency learners generally used visual, aural, and captioned cues simultaneously.

[2]’s ([2]) study, which investigated how 15 Italian lower-intermediate EFL learners responded to captioned videos through interviews, found that students focused more on reading the captions than on listening. The study revealed that low-proficiency students became overwhelmed by the combination of visual, aural, and textual cues, leading them to rely heavily on reading the captions. The simultaneous presentation of visual, aural, and textual information impeded comprehension. This echoed the findings of [22] ([22]), who examined 26 second-semester US university Russian learners (high-level beginners) through questionnaires. The questionnaire results indicated that learners paid the most attention to captions, followed by the video and then the audio. Some participants had difficulty in attending to all three modalities. In the third study, [29] ([29]) used eye tracking and interviews to determine the total amount of time English-speaking learners of Chinese, Arabic, Russian, and Spanish spent on two documentaries dubbed and captioned in the target language. The results showed that these low-intermediate learners primarily focused on the captions, spending an average of 68% of their time fixating on the caption area when captions were present. Spanish and Russian learners exhibited different caption-reading behaviors compared to Arabic and Chinese learners, with fixation times of 63% and 67% for Spanish and Russian learners, respectively, and 75% and 68% for Arabic and Chinese learners. Notably, Chinese learners spent 62% of their time looking at captions when the video content was familiar, compared to 74% when the content was unfamiliar. The Chinese learners reported spending most of their time reading the captions because the Chinese characters were difficult to process. [29] ([29]) also mentioned that some native language studies investigated how readers read still pictures with captions (e.g., [9]; [19]; and [24]). The overall picture emerging from these studies has shown that readers tend to look briefly at the picture first (as few as three eye fixations), the caption next, and then the picture again.

In [18]’s ([18]) multi-proficiency study, which gathered data through observation, verbal report, and interview, the less-proficient learners also relied more on the captions. Two students even mentioned “as if they were doing a reading task instead of listening” ([18]). However, the more proficient learners used captions less often because they were able to follow the sound. They only turned to the captions when they experienced comprehension difficulties. In the other multi-proficiency study, [23] ([23]) examined 35 US Spanish learners of intermediate and upper-intermediate level. In their reflective written protocols, some first-year students reported that they found the captions distracting and had difficulty using image, sound, and captions simultaneously. Experienced learners reported they could use three channels together while watching captioned videos.

These five studies indicate that the way foreign language (FL) learners process captioned videos is closely related to their proficiency levels. As we mentioned above, the two main trends that emerged are that less proficient learners tend to mainly read the captions and that more proficient learners generally typically use visual, aural, and captioned cues flexibly. In a recent study, [16] ([16]) investigated the impacts of subtitle language and learners’ foreign language (FL) proficiency on learning from subtitled academic videos. 131 French-speaking students with different English-proficiency levels watched a video under three conditions: English same- language subtitles (SLS), French native language subtitles (NLS), or no subtitles. Using a between-group design, they measured comprehension (retention, inference, and transfer), self-reported cognitive load, and situational interest. Results show no subtitle effects but a significant proficiency main effect: advanced learners outperformed lower-proficiency peers on inference/transfer tasks and reported reduced extraneous cognitive load. This aligns with cognitive load theory, suggesting that higher proficiency mitigates linguistic processing demands, freeing resources for integrating audio, visual, and textual inputs. These findings resonate with the trends above: advanced learners’ lower cognitive load supports multi-modal integration, whereas elementary learners, facing higher load, may prioritize captions as a compensatory strategy. However, an exception was [3]’s ([3]) study, which involved 10 native Chinese speakers with upper-intermediate English proficiency. This finding contradicted the second trend that more proficient learners used multiple channels. Eight participants reported that they “paid most of their attention to the captions” ([3]). The results suggested that teachers should avoid using captions when incorporating videos to enhance L2 learners’ listening skills. Unlike other cross-sectional studies, [3] ([3]) divided 20 participants into a no-captions group and a captions group. Their procedure also differed from other studies. Participants first watched the video in their native language (Chinese) as a scaffold to help them understand the storyline, a method that is uncommon in both research and classroom contexts. After viewing, participants were shown 21 words and phrases from the video script and instructed to “try and listen for these words and the sentences in which they occurred” ([3]). The researchers explained that their study focused on individual words and phrases rather than overall comprehension. This might have subtly encouraged participants to focus more on the captions.

#### 2.2.3. Main Results from the Longitudinal Studies

All three longitudinal studies were conducted by [26] ([26], [27], [28]). Their results were consistent and demonstrated a trend of continuous improvement. All three studies involved proficient learners at the upper-intermediate to advanced level. The results can be categorized into three groups: (1) relatively less proficient students among the high-level learners used captions as a scaffold; (2) proficient learners alternated between sound, image, and captions, using captions to reinforce their understanding; and (3) most learners exhibited changes in their viewing behaviors over time.

In [26]’s ([26]) study, fifteen upper-intermediate to advanced European ESL learners watched authentic TV programs of various genres for one hour per week over nine weeks. The findings indicated that some learners alternated between spoken and caption cues as needed, while others simultaneously utilized spoken, visual, and caption cues, similar to the behaviors of proficient learners in [18]’s ([18]) and [23]’s ([23]) studies. Initially, some learners found the captions distracting, but this effect gradually diminished over time. Only two learners reported a continued need or desire to use captions after nine weeks. They felt that watching uncaptioned TV programs would have encouraged them to listen more attentively and improve their listening abilities. Additionally, some participants struggled with accents and the excessive amount of information, which hindered their engagement and comprehension. The findings also suggested that captioned programs “may be of limited value for low-level learners” ([26]). Eight low-intermediate to advanced Arabic students in the study group found that the captions changed too quickly.

In [27]’s ([27]) study, 18 learners of European foreign languages at the upper-intermediate to advanced level selected from a wide range of films of various genres and watched them at their own pace and control. In this study, Vanderplank observed similar changes in viewing behaviors among the learners, which aligned closely with those identified in his earlier research ([26]). All participants aimed to watch films confidently, and the presence of captions shifted their focus from “viewing for enjoyment” to “conscious learning”. The participants reported using captions when needed. The more advanced participants used captions primarily as a backup when they struggled to understand the speakers. As the study progressed, they gradually reduced their reliance on captions. All participants noted in their diaries that many films would have been difficult to appreciate without captions. However, two lower-level participants reported still relying on captions by the end of the study. The final questionnaires in [27] ([27]) indicated that most learners experienced an overall increase in confidence by the end of the study. Many participants realized that they relied more on reading captions than on listening. As their confidence grew, they used captions flexibly for “reassurance”.

In [28]’s ([28]) study, thirty-six intermediate-advanced learners of French, German, Italian, and Spanish watched authentic movies of various genres. This longitudinal study consisted of two trials. The first trial was the second longitudinal study, involving 18 learners over six weeks, while the second trial, involving 18 participants, was extended to twelve weeks. The findings closely aligned with those of the previous two longitudinal studies and mirrored the outcomes observed in [18] ([18]) and [23] ([23]). The participants tended to read captions at the start of viewing, particularly when encountering unfamiliar accents, rapid dialogue, or content beyond their proficiency level. Eleven participants expressed concerns about becoming too reliant on captions. Nevertheless, their viewing behaviors evolved over time to “maximize the value of the tri-modal input of sound, text, and visuals” ([28]). Initially, only four participants expressed negative views about the value of captions. However, their attitudes shifted from “hostile” to “positive” over time because, without captions, they would not have been able to follow the speech or understand the plots. Most participants noted that their confidence and ability to watch uncaptioned films improved as they watched more films. Additionally, the findings suggested that a recreational mindset “appeared to hinder the potential benefits of captioning in both formal and informal viewing contexts” ([28]).

### 2.3. Summary of Previous Research

Thus far, prior studies have investigated how language learners process captioned videos both in and beyond classroom settings using cross-sectional or longitudinal data. One of the two main findings is that low-proficiency learners mainly read captions, using them as a scaffold to help them understand the videos. This was observed in three cross-sectional studies involving only low-proficiency learners ([2]; [22]; [29]). This was also found among the less proficient learners in [18]’s ([18]) multi-proficiency study. In the other multi-proficiency study, [23] ([23]) found that more first-year students than third-year students struggled to attend to sound, image, and captions simultaneously, though it was unclear how they resolved this issue.

The second main result, emerging from [18]’s ([18]) and [23]’s ([23]) studies, is that higher-proficiency learners flexibly attended to three channels: sound, image, and captions. Learners in [18]’s ([18]) study used captions to reinforce their comprehension when needed. However, [3]’s ([3]) study, which involved 10 native Chinese speakers with proficient English, contradicted the second result, as more proficient learners did not consistently use multiple channels. Eight participants reported that they “paid most attention to the captions” ([3]).

The results of the three longitudinal studies ([26], [27], [28]) can be categorized into three groups. First, less-proficient learners relied more heavily on captions as a scaffold, a finding consistent with [3]’s ([3]) study. Second, proficient learners alternated between sound, image, and captions, using captions to reinforce their understanding, echoing the findings of [18] ([18]) and [23] ([23]). A novel insight from [28] ([28]) advanced the research by showing that most learners’ viewing behaviors evolved over time. They tended to read captions at the start of viewing. At this stage, the participants’ viewing behaviors resembled those observed in [3]’s ([3]) study.

In summary, this review demonstrates that previous studies do not provide a definitive conclusion about how low-proficiency and, particularly, high-proficiency learners process captioned videos. Given the inconclusive and occasionally conflicting findings, further research is needed to draw definitive conclusions.

### 2.4. Research Gaps

#### 2.4.1. Research Techniques

The tools utilized in most of the nine studies were introspective, mainly questionnaires (e.g., [22] and [28]), verbal reports (e.g., [18]), or interviews (e.g., [3]; [23]; [22]; and [26]). Despite their strengths, they are not capable of capturing the temporal fluctuations in cognitive processes. Their retrospective nature of reporting “can reduce their reliability” ([31]). [29] ([29]) used eye-tracking technology to objectively assess participants’ real-time attention distribution between captions and the videos. However, eye-tracking can be used to investigate how learners attend to videos and captions, but not audio. There is a need for measurements that directly indicate cognitive processing during the viewing of captioned videos. Therefore, I need multiple techniques (such as eye-tracking, paused transcription, and interviews) to complement the deficiencies of each technique and to triangulate the findings from each one. The overall picture can give us a more accurate and objective description regarding how learners pay attention to sound, image, and captions when watching captioned videos.

#### 2.4.2. Basis for Selecting Videos

In some studies, the basis for the selection of videos was not specific, and criteria were too random. However, whether the videos’ difficulty level matches participants’ proficiency is a major limitation to researching how second or foreign language learners process captioned videos. Undoubtedly, if the content difficulty level is too far above participants’ proficiency, they tend to read the captions more, while according to [10]’s ([10]) comprehensible input theory, learners should be provided with input a bit beyond their current level of competence. Therefore, the selected videos should not be so challenging that learners are driven to read the captions. Additionally, other factors which should be taken into consideration include topic familiarity, delivery speed, speaker clarity, accent familiarity, lexical–syntactic difficulty, the correspondence between the visual, spoken, and caption cues, and especially, the level and type of visual movements (talking-head or multi-speaker video) ([33]). One or two voices in a talking-head video do not pose a major challenge to both groups, but in videos “where there are three or more voices, the demands upon the listener are inevitably stepped up” ([7]).

#### 2.4.3. Targeted EFL Learner Groups

The existing research primarily focused on full-caption videos’ viewing behaviors of learners whose mother tongues were mostly Indo-European languages. Thus, further research is needed to know the viewing behaviors of learners in an EFL context in which there exist distinct L1–L2 orthographic, phonological, and semantic differences, such as native Mandarin-speaking Chinese EFL learners. As noted by [29] ([29]), “it is, therefore, worthwhile to investigate whether languages with different writing systems, phonological features and levels of cognitive demand differentially affect language learners’ use of captions” (p. 257).

### 2.5. Research Questions

To address the gaps identified above, the present study focuses on the following two research questions.
Do Chinese students of high and low proficiency primarily focus on listening to the audio or reading captions when watching captioned videos?Do they rely more on captions due to increased visual complexity in multi-speaker videos compared to single-speaker videos?

## 3. Methodology

### 3.1. Participants

The study involved 64 Chinese students from a university in Dalian, Liaoning Province, China. The participants were students from non-English disciplines, such as software engineering, bioengineering, communications, and economics. They had learned English in traditional classrooms, where the focus was mainly on vocabulary, grammar, and reading, with the aim of improving test performance rather than communicative skills.

While a formal power analysis was not conducted due to the exploratory nature of this study examining attention allocation patterns, the sample size was determined by adhering to the widely cited guideline from [11] ([11]), which recommends a minimum of 30 participants per group for detecting medium-sized effects in between-group comparisons. After excluding participants with incomplete or low-quality data, the final sample included 31 low-proficiency (A1/A2 CEFR) and 33 high-proficiency (C1/C2 CEFR) learners.

The low-proficiency group consisted of second-year students (mean age = 19.9, *SD* = 0.3), while the high-proficiency group included third- and fourth-year students (mean age = 20.3, *SD* = 0.47). All participants indicated that they had normal or corrected vision and reported no hearing impairments. The experiment was conducted with the written consent of all participants, who each received CNY 100 (approximately USD 14) as compensation.

### 3.2. Materials

Each group (low proficiency and high proficiency) watched two videos appropriate to their proficiency level: one single-speaker video and one multi-speaker video. The choice of these video types, as opposed to selecting documentaries as [29] ([29]) did, was based on their closer resemblance to real-life communication scenarios.

The low-proficiency group watched an 18 min single-speaker video on “Traffic, Weather, and Pets”, while the multi-speaker video for this group focused on job hunting. The high-proficiency group watched a 15 min single-speaker video on “Teenagers, Adulthood, and Procrastination”, while the multi-speaker video was an excerpt from a US sitcom.

The videos were selected based on criteria suggested by [33] ([33]), including topic familiarity, delivery speed, speaker clarity, accent familiarity, lexical–syntactic difficulty, and the correspondence between visual, spoken, and caption cues, with particular attention to the level and type of visual movements. Two instructors independently assessed the videos for their appropriateness in terms of difficulty and agreed that the videos were well-matched to the proficiency levels of the students. Despite inherent challenges in video selection, this study systematically controlled key variables (e.g., linguistic complexity) to minimize confounding factors, thereby strengthening the methodological rigor of the design.

### 3.3. Instrument: Paused Transcription

In paused transcription, a long recording is paused at pre-arranged irregular intervals. Each pause is followed by a black screen that lasts 10–15 s, during which learners write down the final 4–5 words of the sentence they have just heard. This approach can tap into the learners’ decoding processes and allows researchers to obtain more immediate feedback on how L2 listening is processed ([17]). This is based on psycholinguistic evidence from [8] ([8]), which suggests that listeners can briefly retain a verbatim record of the words they hear until the onset of the following clause ([6]). This technique was used by [6] ([6]) and [30] ([30]) to investigate L2 English learners’ decoding of content words and function words. Additionally, [32] ([32]) employed this method to examine the influence of formulaic language on L2 listener decoding. The validity of the paused transcription technique was analyzed and confirmed by [17] ([17]), who demonstrated its effectiveness in capturing L2 learners’ real-time decoding processes.

In this study, the paused transcription process largely followed the methods used by [6] ([6]) and [30] ([30], [32]). Each captioned video was paused at random natural breaks, and participants transcribed the last 4–5 words of the sentence they had just heard. Each video included 20 random pauses. Of these, 10 captions matched the audio, while intentional errors were introduced into the remaining 10 captions. The underlying assumption was that individuals who relied on captions would primarily consider the content they had read as their reference. Consequently, when encountering misleading captions, they would transcribe what appeared on the screen. In contrast, those who could accurately transcribe despite the mismatch between on-screen captions and auditory input could be considered less dependent on captions.

As mentioned above, participants only needed to transcribe the last 4–5 words of the sentence they had just heard. This was also the reason why I did not explicitly test their working memory. According to [15]’s ([15]) “The Magical Number Seven, Plus or Minus Two”, the typical capacity of working memory for adults is around 7 ± 2 units of information. Given this well-established concept, I assumed that asking participants to recall only a few words from the end of a sentence would not exceed their typical working memory capacity. Therefore, this task was designed more to assess language processing rather than directly measuring working memory performance.

The transcription sections were carefully selected and edited based on several considerations: (1) Pauses were arranged irregularly at natural breaks, primarily at sentence endings, to prevent predictability and maintain video continuity. (2) The duration was set according to participants’ proficiency: 15 s for the low-proficiency group and 10 s for the high-proficiency group, allowing for the transcription of approximately six words. (3) Each pause was inserted after a segment of at least 10 s of continuous playtime to avoid affecting viewing and understanding. (4) Participants only needed to write down the last 4–5 words to avoid extra cognitive load. (5) Edited captions (or misleading captions) were subtle and still made sense both grammatically and contextually without hindering understanding. (6) A black screen followed each misleading caption during transcription, and a countdown reminder played three seconds before the screen disappeared to signal video resumption.

There are four strategies for modifying captions: adding extra words, dropping words, replacing words, and misplacing words. In this study, the misleading captions were mainly created by replacing words, adding extra words, and deleting words.
(1) Adding wordsOriginal ContentMisleading captionsI think everyone should try to walk more.I think everyone should try to walk out more.I don’t have time right now.I don’t have much time right now.Taking control of your lifeTaking control of your own life(2) Deleting wordsOriginal ContentMisleading captionsShe lived with us for over 10 years.She lived with us for 10 years.you need to tell them the truthyou need to tell the truthThis is one technique that you can employ in your own studies.This is one technique that you can employ in your studies.(3) Replacing words
Original ContentMisleading captionsthe subway stations are dirty and a little scary.the subway stations are dirty and a little scaring.They were smart and fun.They were smart and funny.You have to take turns.You ought to take turns.

To determine whether learners relied more on listening to the spoken audio or reading the captions, I counted students’ written transcriptions of video segments containing edited captions. A transcription aligned with the edited captions or the audio awarded 1 point. For example, if students accurately transcribed “They were smart and fun” (matching the original correct version from the audio) instead of the misleading captions “They were smart and funny”, they received 1 point for listening; if students wrote “They were smart and funny” (matching the misleading captions) instead of the original correct version from the audio “They were smart and fun”, they received 1 point for reading. Minor spelling mistakes were disregarded. For instance, if students transcribed the edited caption “They were smart and funny” as “They were smart and funy” (missing one letter ‘n’), they would still receive 1 point for reading because the key criterion here was whether they included the “y” from the edited caption that was not present in the audio, and spelling errors did not affect scoring in this context. However, if learners’ transcriptions matched neither the edited captions nor the original audio, they would be marked as “wrong”. For example, if the speaker in the video said “I don’t have time now” while the edited caption displayed “I don’t have much time now”, but learners wrote “I don’t have any time now”, this would be marked as “wrong”. However, such cases were relatively rare and had little impact on addressing the two research questions, so they were excluded.

### 3.4. Data Collection

The paused transcription test was conducted in a tiered lecture hall at a university. This choice was made for three reasons. First, one of the research objectives was to find an efficient, user-friendly, and scientifically sound method for teachers to assess students’ reliance on captions. This would enable teachers to select suitable captioned videos and decide when or if to remove captions. Therefore, a tiered lecture hall, accessible to all teachers, was the best option. Second, this approach allowed for the simultaneous testing of all participants, increasing efficiency compared to individual testing. Third, this method was cost-effective because the lecture hall was equipped with a projector, a sound system, and a large screen.

Participants first filled out a background questionnaire, which included demographic information (name, gender, age, school, major, and grade), as well as details about their academic and language background, including the length of English study, their English score in the College Entrance Examination, and their scores on standardized tests (CET 4/CET 6). This was followed by an example and instructions explaining the order of the tasks. At the bottom of the first page was an informed consent form, which informed students that the data from the experiment would be used solely for scientific research. Students were required to sign it.

Before the experiment, participants were explicitly instructed to prioritize listening over captions. This was to encourage participants to listen without relying on captions, as they would in real-life interactive situations. They were instructed to write down only what they had heard.

Before the experiment officially began, an example video was shown to demonstrate the procedure and ensure that everyone understood the process. A paused transcription test was then administered. Participants first watched an English talking-head video with English captions. Each captioned video was paused at natural breaks. A black screen followed each pause, and participants transcribed the last 4–5 words they had just heard. Participants then watched the English multi-speaker videos. The process was the same as watching the single-speaker videos. Then, students were free to leave.

## 4. Results

Transcriptions were scored in accordance with the criteria outlined in the research instrument section. The quantitative data were analyzed using SPSS 29. The Shapiro–Wilk test was applied to assess whether the transcription scores (reading and listening) for each group (high/low proficiency) and video type (single/multi-speaker) followed a normal distribution. Based on that, either parametric tests (such as the *t*-test) or non-parametric tests (like the Wilcoxon signed-rank test) were employed to compare two sets of data and ascertain whether a significant difference existed between them. Moreover, the effect size was considered. Cohen’s *r*, a frequently used effect-size metric, particularly suitable for non-parametric data, was utilized (*r* = 0.10 as a small effect, *r* = 0.30 as a medium effect, and *r* = 0.50 as a large effect).

### 4.1. Findings of Research Question 1

The results of the Shapiro–Wilk test revealed that, for the low-proficiency group, both reading scores (*W* = 0.914, *p* = 0.016) and listening scores (*W* = 0.873, *p* = 0.001) significantly deviated from normality (*p* < 0.05 for both). Similarly, the high-proficiency group showed non-normal distributions for reading (*W* = 0.876, *p* = 0.002) and listening (*W* = 0.912, *p* = 0.013). Given the non-normal distribution of the data, non-parametric tests were deemed appropriate for further analysis. To examine whether students of high and low proficiency relied more on reading captions or listening to audio during video viewing, a Wilcoxon signed-rank test was selected to compare the paired reading and listening scores within each group. Effect sizes were calculated using Cohen’s *r* to quantify the magnitude of differences observed.

Transcription scores for listening and reading in single- and multi-speaker videos (total score: 20 points) were compared. Descriptive statistics for the low-proficiency group’s transcription scores revealed a marked preference for reading captions over listening to audio. As shown in Table 1, the mean reading score (*M* = 10.52, *SD* = 3.24) was substantially higher than the mean listening score (*M* = 5.81, *SD* = 2.94). A Wilcoxon signed-rank test confirmed this difference was statistically significant (*Z* = −4.55, *p* < 0.001), with a large effect size (Cohen’s *r* = 0.82), indicating that low-proficiency learners relied overwhelmingly on captions during video viewing, as evidenced by the fact that the median reading scores (11) were double the median listening scores (5), indicating their strong preference for captions.

Conversely, high-proficiency participants demonstrated the opposite pattern. As presented in Table 1, their mean listening score (*M* = 12.27, *SD* = 3.12) out of 20 points significantly surpassed the mean reading score (*M* = 6.70, *SD* = 3.84). A Wilcoxon signed-rank test indicated a statistically significant difference between listening and reading scores (*Z* = −5.12, *p* < 0.001), with an equivalently large effect size (Cohen’s *r* = 0.89). Additionally, the median listening score (11) was nearly double the median reading score (6), further indicating a strong preference for auditory input. These findings suggest that high-proficiency learners prioritized auditory input over captions during video viewing.

These divergent patterns across proficiency levels provide robust evidence that language proficiency moderates learners’ reliance on captions versus audio. Low-proficiency students depended significantly on captions (median reading score = 11 and median listening score = 5; reading > listening, *Z* = −4.55, *p* < 0.001, *r* = 0.82), whereas high-proficiency students prioritized audio (median reading score = 6 and median listening score = 11; listening > reading, *Z* = −5.12, *p* < 0.001, *r* = 0.89). These findings align with theoretical frameworks positing that lower proficiency necessitates scaffolding via textual support, while higher proficiency enables integrated audio–visual processing.

### 4.2. Findings of Research Question 2

The second research question examines whether learners rely more on captions in multi-speaker videos than in single-speaker videos due to the increased visual complexity.

The normality of the reading and listening scores for both single-speaker and multi-speaker videos was assessed using the Shapiro–Wilk test. For the low-proficiency group, reading scores in single-speaker videos (*W* = 0.901, *p* = 0.007) and multi-speaker videos (*W* = 0.892, *p* = 0.004) significantly deviated from normality. Similarly, for the high-proficiency group, reading scores in both single-speaker (*W* = 0.884, *p* = 0.003) and multi-speaker videos (*W* = 0.891, *p* = 0.005) were non-normally distributed. These results (*p* < 0.05 for all tests) violated the assumptions of parametric tests, necessitating non-parametric analyses. For between-video comparisons, additional Wilcoxon signed-rank tests were conducted to evaluate caption reliance in multi-speaker versus single-speaker videos. Effect sizes were calculated using Cohen’s *r*, interpreted as small (*r* = 0.10), medium (*r* = 0.30), or large (*r* = 0.50) effects based on established thresholds.

To address this question, I calculated and compared the reading scores between single-speaker and multi-speaker videos within each proficiency group. To be specific, two statistical tests were conducted: (a) For the low-proficiency group, the transcription scores for reading from both single-speaker and multi-speaker videos were calculated and compared to determine if there was a significant difference. (b) Similarly, for the high-proficiency group, the transcription scores for reading from the two types of videos were calculated and compared.

Table 2 revealed low-proficiency learners’ higher reading scores in multi-speaker videos (*M* = 6.06, *SD* = 1.93) compared to single-speaker videos (*M* = 4.94, *SD* = 1.84). A Wilcoxon signed-rank test confirmed this increase was statistically significant (*Z* = −4.21, *p* < 0.001), with a large effect size (*r* = 0.75). This suggests low-proficiency learners relied more on captions when processing multi-speaker content. The median reading score increased from five (single-speaker) to six (multi-speaker), reflecting a consistent upward shift in typical caption reliance. Despite no change in the median for high-proficiency learners (three → three), high-proficiency learners, while still prioritizing listening overall, showed a small but significant increase (*p* = 0.003) in caption reliance during multi-speaker videos (*M* = 3.70, *SD* = 1.89) compared to single-speaker videos (*M* = 3.09, *SD* = 1.97), with a moderate practical effect (*r* = 0.52).

This suggests that increased visual complexity (multi-speaker content) amplifies caption reliance across proficiency levels, but the impact is modulated by learners’ language skills. Low-proficiency learners depend more heavily on captions while high-proficiency learners showed a small but significant increase in caption reliance while watching multi-speaker videos.

## 5. Discussion

The findings of this study align with, and in some cases extend, the existing literature on how learners of different proficiency levels process captioned videos, offering subtle insights into the interplay between proficiency levels, cognitive load, and instructional design.

### 5.1. Interpretation of Findings

Consistent with the cross-sectional studies ([2]; [22]; [29]), low-proficiency participants in this study demonstrated a strong reliance on captions with reading scores significantly surpassing listening scores (Z = −4.55, *p* < 0.001), which supports the notion that low-proficiency learners use captions as a scaffold to aid comprehension. The results echo [2]’s ([2]) observation that low-proficiency learners often become overwhelmed by the tri-modal input, leading them to focus primarily on reading captions. Similarly, [22]’s ([22]) finding that low-proficiency learners prioritize captions over audio and visual cues is reinforced by the current data.

In contrast, high-proficiency participants in this study demonstrated superior auditory processing (listening > reading, Z = −5.12, *p* < 0.001), reflecting their ability to allocate cognitive resources efficiently. This finding aligns with [18]’s ([18]) and [23]’s ([23]) studies, which found that higher-proficiency learners are more adept at flexibly integrating visual, auditory, and textual cues. However, the slight increase in incongruent captions when watching multi-speaker videos suggests that even advanced learners may occasionally rely on captions due to increased cognitive load, such as when tracking multiple speakers. This nuanced finding adds to the literature by highlighting that proficiency level is not the sole determinant of reliance on captions; situational factors, such as video complexity, also play a role. The results also resonate with [26]’s ([26], [27], [28]) longitudinal studies, which found that learners’ reliance on captions evolves over time. While low-proficiency learners in this study consistently relied on captions, high-proficiency learners demonstrated a more balanced use of auditory and textual cues, similar to the proficient learners in Vanderplank’s studies who used sound, image, and captions as needed. The current study’s findings suggest that learners’ processing strategies can vary based on video type, even within a single experimental session, echoing Vanderplank’s observation that viewing behaviors evolve with exposure and experience.

However, the results contrast with [3]’s ([3]) study, which found that even high-proficiency learners relied heavily on captions. This discrepancy may be attributed to differences in experimental design. Chai and Erlam’s study focused on word and phrase recognition, which may have inadvertently encouraged participants to prioritize captions. In contrast, the current study employed a more naturalistic approach, enabling participants to process captioned videos holistically. This methodological difference underscores the importance of task design in shaping learners’ reliance on captions.

Visual complexity, operationalized through multi-speaker videos, increased caption reliance among low-proficiency learners (reading scores increased by 23.7%, *Z* = −3.85, *p* < 0.001). This aligns with CLT’s prediction that heightened intrinsic load (e.g., tracking multiple speakers) amplifies dependence on extraneous supports like captions. High-proficiency learners, however, exhibited only a modest increase in caption reliance (19.7%, *Z* = −2.97, *p* = 0.003), suggesting their superior ability to parse complex auditory–visual streams without over-relying on text. This divergence underscores proficiency as a critical moderator of cognitive load management.

### 5.2. Implications: Adaptive Strategies for Captions and Videos

The findings of this study suggest tailored strategies to adjust multimedia inputs for learners at different proficiency levels. These strategies aim to balance cognitive load while fostering skill progression, directly addressing the observed differences in caption reliance and auditory processing.

In practice, educators could design tiered caption removal protocols to align with proficiency thresholds and phase out captions dynamically. For example, low-proficiency learners can start with full captions, then highlight keywords (e.g., idioms and complex grammar) while dimming others, and finally progress to no captions—a progression that nudges learners toward selective attention as proficiency improves. High-proficiency learners can be challenged with partially captioned or uncaptioned materials to foster listening skill development. Also, educators may consider calibrating video complexity by prioritizing single-speaker content for low-proficiency learners to help mitigate cognitive overload, while multi-speaker videos could be reserved for advanced learners as potential strategic listening challenges.

In addition, instead of a one-size-fits-all approach, teachers can use simple tools (like adjustable caption opacity in PowerPoint or YouTube) to decrease opacity from 100% to 30% and phase out captions in sync with student progress, forcing a strategic shift from text-dependent decoding to auditory-focused processing. For instance, the progress could start with full captions in Week 1 to build learner confidence, followed by reducing caption opacity to 50% while highlighting keywords in Week 3 to train selective attention. By Week 6, learners would transition to no captions for familiar topics, gradually fostering auditory independence through carefully calibrated challenges.

Finally, to address the challenge of multi-speaker scenes overwhelming beginners (as shown in the data), this study advocates for a complexity-aware design framework that tags videos with “complexity scores” (e.g., 1–5 stars) and pairs them with adaptive caption support. For instance, a five-star debate video, characterized by fast-paced verbal exchanges and diverse voices, would retain highlighted keywords to scaffold comprehension, while a two-star monologue with clear speech could be presented without captions to encourage auditory engagement. This approach leverages visual cues to dynamically balance cognitive load, allowing learners to progressively tackle more challenging content while maintaining focus on critical linguistic elements. By aligning multimedia complexity with tiered support systems, educators can systematically guide learners from reliance on textual aids to independent audio processing, fostering skill development through intentional difficulty calibration.

### 5.3. Limitations and Future Directions

While this study provides insights into how low- and high-proficiency learners allocate their attention and how caption types influence learner viewing behaviors, several limitations should be acknowledged alongside critical questions for future investigation.

First, I recognize that uncontrolled variables—including video content characteristics (e.g., humor) and learner factors (individual interest and prior knowledge)—may have mediated the observed effects. The video selection process prioritized linguistic appropriateness (grammar structures, lexical difficulty, and speech rate), matching participants’ proficiency levels, followed by secondary controls for accent consistency, topic familiarity, and comparable duration. While this approach ensured pedagogical suitability, it necessarily constrained my ability to perfectly balance other potentially influential factors, particularly stylistic elements and nuanced topic differences between single-speaker and multi-speaker videos. Future research could implement stricter content-matching protocols and include measures of individual differences to better isolate the specific effects of caption variations. These refinements would help disentangle the observed effects from potential confounders while preserving the ecological validity of authentic video materials.

Second, the transcription-based measurement, while providing direct evidence of attention allocation between audio and captions, cannot capture more subtle cognitive processes like the depth of processing or comprehension. Although eye-tracking technology alone cannot distinguish whether learners are actively reading captions or listening to the audio, its incorporation could synergistically address the second research question. Future research could adopt an innovative methodology integrating eye-tracking technology with a modified paused transcription task. Specifically, rather than verbatim text transcription, participants would engage in answering multiple-choice questions during pre-designated video pauses while maintaining head stability for eye-tracking calibration. This hybrid approach would enable researchers to systematically examine how learners dynamically allocate cognitive resources across multiple modalities (audio, visual, and captions) during multimedia engagement.

Third, the data in this study do not provide a definite answer to what difficulties are desirable in second/foreign language learning. Specifically, the observed preferences for modality-specific attention, whether toward captions or audio, reflect strategic behaviors rather than confirmed learning benefits. To address this gap, future research could explore this concept further. For instance, a follow-up study might examine how the development of listening abilities varies between low- and high-proficiency learners when exposed to captioned versus non-captioned materials. Such an investigation would help determine two critical issues: (1) whether removing captions or reducing caption support truly benefits high-proficiency learners, and (2) whether low-proficiency learners could likewise gain advantages from caption removal under specific conditions. Importantly, these inquiries align with the findings of [35] ([35]), who cautioned that learner preferences for fluent conditions (e.g., full captions) do not necessarily correlate with optimal learning outcomes. In their study, moderate discrepancies between narration and on-screen text in multimedia learning were found to create desirable difficulties, even though learners consistently preferred on-screen text to match the audio narration.

Ideally, future research on captions should not only analyze strategic differences in caption usage among learners at varying proficiency levels, but also focus on developing dynamic captioning systems that leverage eye-tracking and AI-driven proficiency analytics for real-time adaptation. These systems would automatically adjust caption presentation based on learners’ real-time proficiency (e.g., listening comprehension and vocabulary mastery), guiding them to adopt the most contextually appropriate caption strategies for their current learning stage. This aligns with the perspective articulated by [12] ([12]), who emphasized that future research should aim not only to identify individual learners’ strategy differences for outcome prediction, but more importantly, to systematically design interventions that encourage the adoption of contextually optimal strategies across different proficiency levels.

## 6. Conclusions

In conclusion, this study provides further evidence that proficiency level significantly influences how learners process captioned videos. Low-proficiency learners predominantly rely on captions as a scaffold for comprehension, while high-proficiency learners demonstrate a greater ability to integrate auditory and textual cues. Additionally, video complexity (the presence of multiple speakers) can increase reliance on captions, particularly for low-proficiency learners.

This study contributes to the cognitive applications in education by operationalizing “desirable difficulties” ([1]) in multimedia instruction. For low-proficiency learners, captions serve as a scaffold that balances challenge and support. Conversely, high-proficiency learners’ preference for auditory input (as evidenced by higher listening scores) suggests that reducing caption reliance may introduce beneficial difficulties, potentially enhancing real-time auditory processing. The findings also demonstrate that contextual factors (e.g., video complexity) dynamically modulate the balance between challenge and support. Future research should investigate the development of listening abilities across learners of varying proficiency levels under different captioning modalities, while also exploring dynamic systems that adapt captions in real time based on learners’ evolving linguistic profiles and contextual demands.

## Figures and Tables

**Table 1 behavsci-15-00542-t001:** Listening vs. reading transcription scores for both groups and Wilcoxon signed-rank test results.

Group	Transcription Scores for Listening (*M*, *SD*)	Median	Transcription Scores for Reading (*M*, *SD*)	Median	*Z*	*p*	Cohen’s *r*
Low (n = 31)	5.81 (2.94)	5	10.52 (3.24)	11	−4.55	<0.001	0.82
High (n = 33)	12.27 (3.12)	11	6.70 (3.84)	6	−5.12	<0.001	0.89

**Table 2 behavsci-15-00542-t002:** Comparison of reading scores for both videos and Wilcoxon signed-rank test results.

Group	Single-Speaker (Reading)	Median	Multi-Speaker (Reading)	Median	*Z*	*p*	Cohen’s *r*
Low (n = 31)	*M* = 4.94, *SD* = 1.84	5	*M* = 6.06, *SD* = 1.93	6	−4.21	<0.001	0.75
High (n = 33)	*M* = 3.09, *SD* = 1.97	3	*M* = 3.70, *SD* = 1.89	3	−2.97	0.003	0.52

## Data Availability

The data presented in this study are available on request from the corresponding author due to privacy.

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
