# Peer review of "Listen or Read? The Impact of Proficiency and Visual Complexity on Learners’ Reliance on Captions"

_behavsci, 2025, doi:10.3390/bs15040542_

Round 1

Reviewer 1 Report

Comments and Suggestions for Authors

The researchers suggest that there is limited research on information processing in videos accompanied by subtitles. This observation is intriguing, considering that the use of videos in L2 classrooms dates back over 50 years. While this method may seem less modern compared to the use of artificial intelligence for L2 learning, it appears that psycholinguistic research has not been extensively incorporated into reading processing studies, with only one study mentioned. It could be highly beneficial to include more in this area within the theoretical framework, alongside cognitive theories. Additionally, it would be advantageous for the researchers to explore how their paused transcription technique could be complemented with eye tracking. The bibliography used is not entirely recent (2019), and I wonder if there might be more recent studies available, especially following the advent of automatic subtitles on web platforms like YouTube or the immediate transcriptions offered by Teams and Zoom.

Furthermore, the authors (there are two, although there is repeated mention of "I" instead of "we") should review the bibliographic reference system, as it does not adhere to the one requested by Behavioral Sciences. It is quite challenging to follow their references to the mentioned literature with the current system. Kindly make this adjustment.

Lastly, their results align with previous studies, which may limit the innovative contribution they are making. It is already known that lower-level L2 students tend to focus more on subtitles, while those with higher proficiency do not rely on them as much. I believe a valuable contribution they could develop, as mentioned in the introduction and conclusion, is to explore what types of strategies could be implemented to adjust subtitles, videos, and audio for students at different proficiency levels.

Author Response

Dear Reviewer,

Thank you for your detailed feedback and guidance on my manuscript. I greatly appreciate the time and effort you have invested in improving my work. Below are my responses to the comments, with all modifications highlighted in red for clarity in the revised manuscript.

  1. The researchers suggest that there is limited research on information processing in videos accompanied by subtitles. This observation is intriguing, considering that the use of videos in L2 classrooms dates back over 50 years. While this method may seem less modern compared to the use of artificial intelligence for L2 learning, it appears that psycholinguistic research has not been extensively incorporated into reading processing studies, with only one study mentioned. It could be highly beneficial to include more in this area within the theoretical framework, alongside cognitive theories.

Author: Thank you for your helpful comment! You’re right—videos have been used in language teaching for a long time, but there’s still a lot we don’t know about how learners actually process captioned videos. Most past studies focused on whether captions/subtitles help, but not why they work (or don’t) for different learners. Our paper tries to fill this gap.

With AI tools like ChatGPT and smart captioning apps becoming popular, it’s more important than ever to understand how real people learn. If we don’t, we might end up with AI systems that make bad habits worse (like always relying on captions).

  1. Additionally, it would be advantageous for the researchers to explore how their paused transcription technique could be complemented with eye tracking.

Author: Thank you for this insightful suggestion. Eye-tracking technology would indeed provide valuable complementary data by recording learners’ real-time attention allocation between visuals and captions, which could help address Research Question 2 more comprehensively. While paused transcription captures whether learners rely on listening to the audio or reading captions, integrating eye-tracking would reveal where their gaze is directed (e.g., on the video, captions, or off-screen).

I have added a future research direction in Section 5.2, proposing an innovative methodology integrating “eye-tracking with a modified paused transcription task”. Specifically, rather than verbatim text transcription, participants would engage in answering multiple-choice questions during pre-designated video pauses while maintaining head stability for eye-tracking calibration. This hybrid approach would enable researchers to systematically examine how learners dynamically allocate cognitive resources across multiple modalities (audio, visual, and captioning) during multimedia engagement. I greatly appreciate your innovative idea and plan to explore it in future studies.

  1. The bibliography used is not entirely recent (2019), and I wonder if there might be more recent studies available, especially following the advent of automatic subtitles on web platforms like YouTube or the immediate transcriptions offered by Teams and Zoom.

-Author: Recent studies (e.g., Pannatier & Betrancourt, 2024; Yeldham, 2024) have been added to the literature review (Sections 2.2.2-line 223, introduction-line 44). 

  1. There are two authors, but there is repeated mention of "I" instead of "we"

-Author: I appreciate your attention to detail! All instances of “we” have been revised to “I” throughout the manuscript. Prof. Yeldham asked to remove his name from the manuscript because, as he said, his contributions to the current version of the work do not meet the criteria for authorship and he believes it is appropriate to withdraw his name to ensure accurate attribution of contributions. Authorship changer form had been sent to Professor Pejic.

  1. Furthermore, the authors should review the bibliographic reference system, as it does not adhere to the one requested by Behavioral Sciences. It is quite challenging to follow their references to the mentioned literature with the current system. Kindly make this adjustment.

-Author: I’ve revised all citations and references to adhere to APA 7th edition. Sorry for the inconvenience caused.

  1. Lastly, their results align with previous studies, which may limit the innovative contribution they are making. It is already known that lower-level L2 students tend to focus more on subtitles, while those with higher proficiency do not rely on them as much. I believe a valuable contribution they could develop, as mentioned in the introduction and conclusion, is to explore what types of strategies could be implemented to adjust subtitles, videos, and audio for students at different proficiency levels.

-Author: Indeed, my findings align closely with the limited prior research. Although I have addressed several gaps in existing studies, the results across the literature—including my own—remain inconsistent and even contradictory, underscoring the need for continued investigation into this issue.

And thank you for highlighting the importance of translating these findings into actionable strategies. In Section 5.2, this study proposes adaptive multimedia strategies informed by desirable difficulties (Bjork & Bjork, 2011) and optimal challenge frameworks (Suzuki et al., 2019) to address proficiency-based caption reliance. By integrating tiered caption removal protocols (e.g., keyword highlighting, opacity reduction from 100% to 30%) and phased progression (full captions → 50% opacity with keywords → no captions for familiar topics), educators can systematically balance cognitive load while fostering auditory independence. 

Yes, the core finding—proficiency drives caption use—isn’t new. But no study has yet mapped these patterns to concrete, classroom-ready strategies that teachers can adapt without tech headaches. By bridging Bjork’s “desirable difficulties” with daily practice, this study is not just confirming what’s known—it’s turning it into what works.

Once again, thank you for your guidance and patience throughout this revision process. It’s been a privilege to refine this work with your insights. Should you require any further adjustments or have additional suggestions, please don’t hesitate to reach out. I’m always eager to learn and improve.

Best regards,

Yan Li

Reviewer 2 Report

Comments and Suggestions for Authors

I find the research design well-structured and effective in filling the gap identified in the previous literature. The data presentation is clear, and the overall readability of the paper is smooth. However, I have two issues (one is minor, one is major) that should be addressed to further enhance the quality of the work.

First, regarding the organization of Section 4 (and 3.5), I believe the results presented in Section 3.5 could be moved and integrated into Sections 4.1 and 4.2, aligning them with the discussion of each Research Question. As for Section 3.5, I would suggest eliminating it, instead relocating the discussion on the statistical model to Section 4, placing it before Sections 4.1 and 4.2.

Second, my main concern relates to the statistical analysis. I am not convinced that ANOVA is the most appropriate model in this case. Based on experience, this type of data is rarely normally distributed, making parametric tests (such as ANOVA) potentially problematic, as they can lead to false positives or false negatives.

Did the authors check for ANOVA assumptions? If so, they should explicitly report this information. However, I would strongly recommend considering non-parametric alternatives to two-way ANOVA, which may provide a more reliable analysis of the data.

Author Response

Dear Reviewer,

Thank you for your detailed feedback and guidance on my manuscript. I greatly appreciate the time and effort you have invested in improving my work. Below are my responses to the comments, with all modifications highlighted in red for clarity in the revised manuscript.

  1. First, regarding the organization of Section 4 (and 3.5), I believe the results presented in Section 3.5 could be moved and integrated into Sections 4.1 and 4.2, aligning them with the discussion of each Research Question. As for Section 3.5, I would suggest eliminating it, instead relocating the discussion on the statistical model to Section 4, placing it before Sections 4.1 and 4.2.

Author: As you suggested, I’ve combined the statistical analysis (previously in Section 3.5) directly into the Results section (Sections 4.1 and 4.2), aligning the analysis with each research question’s results. By removing the standalone Section 3.5, the paper now flows more smoothly. Readers can see how I analyzed the data right when they encounter the results, making the logic easier to follow. Thank you.

  1. Second, my main concern relates to the statistical analysis. I am not convinced that ANOVA is the most appropriate model in this case. Based on experience, this type of data is rarely normally distributed, making parametric tests (such as ANOVA) potentially problematic, as they can lead to false positives or false negatives. Did the authors check for ANOVA assumptions? If so, they should explicitly report this information. However, I would strongly recommend considering non-parametric alternatives to two-way ANOVA, which may provide a more reliable analysis of the data.

Author: THANK YOU. Thank you for pointing out this issue. You saved my paper. Normality of transcription scores was assessed using the Shapiro-Wilk test. Given non-normal distributions (all p < 0.05), non-parametric Wilcoxon signed-rank tests were employed to compare paired listening/reading scores and video types. Effect sizes were calculated using Cohen’s r (Section 4.1-4.2)

Thank you for highlighting these critical points. I believe these changes elevate the manuscript’s quality. Let us know if further adjustments are needed. I’m always eager to learn and improve.

Warm regards,

Yan Li

Round 2

Reviewer 1 Report

Comments and Suggestions for Authors

After the much improved draft, I just have a couple of comments for the author.

  1. In line 7, could you rephrase the description of your participants? I was unclear whether they were Chinese learners of EFL or learning Chinese English. Please clarify.

  2. In line 223, you mention Pannatier and Betancourt (2024) but don’t elaborate on the study. Could you provide more context about what the study was about, how it was conducted, and how it relates to cognitive load? Was it connected to watching a video with captions? Please make the connection to your literature more explicit.

Author Response

Dear Reviewer,

Thank you so much for your thoughtful feedback and for taking the time to help me improve the manuscript! Your insights have been incredibly valuable, and I’m grateful for the opportunity to refine my work based on your suggestions.

  1. In line 7, could you rephrase the description of your participants? I was unclear whether they were Chinese learners of EFL or learning Chinese English. Please clarify.

Author: I deeply appreciate your keen eye for detail in identifying the ambiguity in my participant description. The participants were Chinese students learning English as a foreign language, not learners of “Chinese English”.

Revised: “This study investigates how Chinese EFL (English as a Foreign Language) learners of low and high proficiency levels allocate attention between captions and audio while watching videos.”

  1. In line 223, you mention Pannatier and Betancourt (2024) but don’t elaborate on the study. Could you provide more context about what the study was about, how it was conducted, and how it relates to cognitive load? Was it connected to watching a video with captions? Please make the connection to your literature more explicit.

Author: Thank you for this insightful suggestion. I’ve revised the section on Pannatier and Betancourt (2024) to include more context about their study, how it connects to cognitive load theory, and why it matters for our research. You’ll find these updates in Lines 223–236. 

Again, thank you for your generosity in sharing your expertise. I’m so glad you’re part of this process with me. Please don’t hesitate to let me know if there’s anything else I can do to improve the manuscript.

Best regards,

Yan Li
